# Adaptation and Psychometric Properties of the Scale of Positive and Negative Experience (SPANE) in the General Colombian Population

**DOI:** 10.3390/ijerph18126449

**Published:** 2021-06-15

**Authors:** Marta Martín-Carbonell, Irene Checa, Martha Fernández-Daza, Yadid Paternina, Begoña Espejo

**Affiliations:** 1Psychology Department, Cooperative University of Colombia, Troncal del Caribe S/N, Santa Marta 470002, Colombia; martha.fernandezd@campusucc.edu.co (M.F.-D.); yadid.paternina@ucc.edu.co (Y.P.); 2Department of Behavioral Sciences Methodology, University of Valencia, Av. Blasco Ibáñez, 21, 46010 Valencia, Spain; irene.checa@uv.es (I.C.); bespejo@uv.es (B.E.)

**Keywords:** positive and negative experiences, SPANE, wellbeing, psycho-metric properties, measurement invariance, confirmatory factor analysis, Colombian population, quality of life, psychological assessment, structural equation modeling

## Abstract

(1) Background: Diener’s Scale of Positive and Negative Experiences (SPANE) assesses the presence and intensity of positive and negative affects, since these are considered basic aspects of the study of well-being. This article studies its psychometric properties in the general Colombian population. (2) We conducted a cross-sectional study of a sample of 1255 Colombians and we used structural equation modeling to confirm the bifactor structure. Additionally, we studied invariance by gender, and convergent and concurrent validity. (3) We found acceptable fit indicators for the bifactor model (CFI = 0.889, RMSEA = 0.046, SRMR = 0.059) as well as for the convergent (CFI = 0.909, RMSEA = 0.050, SRMR = 0.063) and concurrent (CFI = 0.966, RMSEA = 0.036, SRMR = 0.041) validity models. We did not confirm total invariance across gender, although we found configural and metric invariance, so percentiles by sex were provided. (4) Conclusions: The SPANE is a valid and reliable measure to assess well-being among the Colombian population, although we alert researchers to the risk of comparing affectivity average scores between sexes.

## 1. Introduction

Research in recent decades shows two great traditions in the study of subjective well-being: hedonic perspectives (related to happiness); and eudemonic perspective (related to optimal development) [1]. From a eudemonic approach, well-being is considered to be linked to the ability to effectively manage one’s environment and feelings of personal growth. From a hedonic approach, well-being is associated with pleasure and happiness; that is, the balance between pleasant and unpleasant emotions [2]. Current research concludes that understanding well-being requires taking into account both approaches (hedonic and eudemonic). In fact, it contemplates three main components in subjective well-being: positive affect (pleasant feelings), negative affect (painful feelings) and satisfaction with life. In turn, this makes up the assessment individuals make when balancing positive and negative affects in their own life, by evaluating how well they are doing, based on their own personal aspirations and goals [3,4].

The importance of the affective component of well-being should not be underestimated, as different studies have proven its value in understanding the role of sociocultural factors [5] in terms of well-being, as well as research that has demonstrated the impact of positive affectivity on health [6]. This affective component of well-being has been conceptualized as people’s assessments of the emotions they experience in their daily lives, such as sadness, fear, anger, joy, etc. [7]. More recently, Diener et al. [8] described the affective component of well-being as a state of predominance of positive affect, with fewer periods of negative affect. The best way to assess the affective component, e.g., in terms of frequency vs. intensity, has been discussed, and most researchers consider that frequency may be a more appropriate measure, since people who experience high levels of well-being rarely feel high-intensity positive emotions, although they regularly feel moderately happy or content [9].

Since most measures used to assess subjective well-being are self-reported, the cornerstone of this research focuses on the study of its psychometric properties [10]. One of the most commonly used scales is the Positive and Negative Affects Schedule (PANAS) [7]. However, its usefulness for measuring the subjective well-being construct has been questioned [11] as it focuses on measuring the affects’ intensity (rather than their frequency). It also uses specific adjectives instead of general ones, and it is further argued that some of the adjectives included in the PANAS scale are not actually affective (e.g., active, strong, alert), and that it does not consider central adjectives to describe the experience of well-being and discontent.

Diener’s Scale of Positive and Negative Experiences (SPANE) [12] assesses the presence and intensity of positive and negative affects, since these are considered basic aspects of the study of well-being, overcoming the limitations found in the PANAS. According to Diener, ‘the SPANE captures positive and negative feelings regardless of their provenance, arousal level, or ubiquity in western cultures where most scales have been created. In this way, our scale can better reflect the full set of feelings felt by individuals around the globe and give them the proper positive and negative weighting. By including labels such as ‘‘good’’ and ‘‘positive’’, and ‘‘bad’’ and ‘‘negative’’, that reflect all types of feelings, the SPANE assesses the full range of possible desirable and undesirable experiences’ ([12], p. 145). Additionally, the SPANE asks about feelings experienced over the last four weeks, which encourages respondents to focus less on the feelings they have at the time of the assessment and more on those related to their self-concept concerning their typical emotional state.

This scale has become popular worldwide due to its ease of application and interpretation, in addition to its powerful conceptual and empirical foundation. It is recommended [13] as one of the most valid measures of emotional well-being. Its author has generously authorized its free use, as long as its copyright is respected.

It is comprised of two subscales: SPANE-P and SPANE-N. The first contains three general items (good, positive, pleasant) and three specific items (happy, joyful, content) that measure positive emotional experiences, whereas the second is made up of three general items (negative, bad, unpleasant) and three specific items (sad, afraid, angry) that measure negative emotional experiences. In both cases, the six general items attempt to describe feelings that people seek or avoid and value or disdain. They can also be applied to a wide range of more specific feelings. In addition, the SPANE’s six specific items express feelings which are traditionally considered relevant when describing experiences related to well-being and discontent [13].

The SPANE is available in several languages and has been validated in different countries, such as Germany [14], Canada [15], India [16], Japan [17], Portugal [18], Turkey [19], Russia [20], Serbia [21,22], Greece [23], Italy [24] and China [25]. Furthermore, it has been adapted by Spanish speakers several times. In Peru, Cassaretto [26] reported having used back translation and studied the factorial structure, reliability, and convergent and divergent validity in a sample of university students. So did Daniel-González in Mexico [27]. In Chile [28], a study was conducted with a sample of high school students using the Spanish translation published on Diener’s website. In Spain, the adaptation to Spanish was carried out by Espejo et al. [29].

Almost all studies corroborated a structure of two inversely related factors (SPANE-P and SPANE-N), and their relationship with personality variables (e.g., neuroticism and extraversion) and alternative measures of well-being, happiness and life satisfaction [14,15,18,30].

As far as we know, there is no validation of the Scale of Positive and Negative Experience (SPANE) in the Colombian population. Taking the recognized advantages of this instrument into account, the purpose of this article is to study its psychometric properties in the general Colombian population. We were specifically interested in corroborating whether the factorial structure is replicated, evaluating invariance by gender, and assessing the convergent and concurrent validity with other measures of subjective well-being (positive and negative affectivity, flourishing and life satisfaction), as well as with optimism and pessimism, based on their link to affectivity. We use the Structural Equation Modeling (SEM) methodology to study validity because it allows us to estimate the error contained in the observed scores, as is done when estimating confirmatory factor analysis (CFA). If the correlation between the total scores of the different measures used to study validity is calculated, correlations between observed scores will be obtained. However, since these observed scores contain errors, these correlations will underestimate the validity. If the correlation between the latent variables is calculated using an SEM, the error (which is estimated separately) will be extracted from these correlations, as occurs when estimating a CFA. For this reason, performing an SEM is a more reliable way to estimate the relationship between measures.

## 2. Materials and Methods

### 2.1. Participants and Procedure

The study was approved by the Ethics Committee of the Cooperative University of Colombia, which guarantees that data collection complies with the Colombian Law on Data Protection, thus ensuring its confidentiality and anonymity.

Two Spanish versions of the SPANE scale were selected, for which the back-translation procedure was described: the one reported by Cassareto Bardales [26] and the one conducted by Espejo et al. [29]. These were evaluated independently by a psychologist experienced in adapting psychological evaluation procedures and by five students who were attending the last semester of a Psychology degree program. Said evaluators reached a consensus about which of the two versions was more appropriate for the Colombian population, choosing the one conducted by Espejo et al. [29] for use in our study because items were better understood. However, some judges raised doubts as to whether the adjective ‘asustado’ [scared] would be understood by Colombians in a similar way to that of people in Spain, or if the adjective ‘temeroso’ [afraid] would be a better fit, so we decided to try both.

Based on the recommendations made by various authors [31,32], this version was piloted with 13 items (including ‘temeroso’ and ‘asustado’) in a qualitative study, in which 14 young volunteers over 18 years participated (Age max = 81, average age = 36.3, SD = 1.5), nine were women and five men, eight had university studies, three had completed high school studies and three only primary studies.

Participants answered the questionnaire in paper and pencil format on an individual basis. In a second session, they were asked to respond to the online version that was used in this study. Finally, a semi-structured cognitive interview was conducted with questions on their understanding of instructions and adjectives, cognitive processes involved when choosing the answers and potential challenges in the response categories, both in the paper and online version.

The cognitive interview allowed us to confirm that the Spanish version was better understood and accepted by the pilot participants, who also claimed not to have perceived differences between the instrument’s two modes of application. They all considered that the adjective ‘temeroso’ (afraid) was more in line with their self-assessment of their typical emotional state compared to the word ‘asustado’ (scared) since that refers to a more intense but fleeting emotion.

Participants were recruited by different means (e-mail, social networks, and through face-to-face application). Data were collected online using Limesurvey, an open-source survey tool. Before starting the survey, informed consent was presented. The participants had to accept in order to continue.

Only the responses of Colombian people of legal age (over 18 years old) were considered. The sample of the study consists of 1255 participants between 18 and 67 years old (Mean = 25.62, SD = 12.1). Of them, 35.8% (*n* = 449) are men, 49.9% (*n* = 538) have university studies, 41.2% (*n* = 517) have completed high-school level, and 13.2% (*n* = 200) have finished compulsory secondary studies or have only primary studies. On the other hand, 75.5% (*n* = 948) are single, 22% (*n* = 274) are married or cohabitating, and 2.5% (*n* = 31) are divorced or widowed. Of the total sample, 43.9% (*n* = 551) are students, 26.1% (*n* = 327) are students with sporadic or part-time jobs, 23.7% (*n* = 298) are employed, 4.9% (*n* = 62) are unemployed, 1% (*n* = 12) are not working nor looking for a job, and 0.4% (*n* = 5) is retired.

### 2.2. Instruments

Scale of Positive and Negative Experience (SPANE) [12]. This scale indicates how individuals evaluate the frequency with which they experience positive and negative feelings, as well as their emotional balance. To do so, 12 adjectives arranged in two subscales of six items each are used: SPANE-P (positive affects) and SPANE-N (negative affects). SPANE has a 5-point Likert scale (from 1-very rarely or never to 5-very often or always). Total scores range from 6 to 30. High scores indicate high positive or negative affect. A balanced measure can be obtained by subtracting the positive and negative total scores, that range from −24 to 24 (MET2).

Positive and Negative Affect Schedule (PANAS) [7]. This has 20 items and uses a 5-point Likert scale (from 1-not at all to 5-extremely). Total scores range from 10 to 50, so high scores indicate high affect. We used the Colombian version, that was preliminarily validated in Colombian women by Cantor Clavijo [33], showing internal consistency scores of 0.71. In this sample, Cronbach’s alpha is 0.814 for positive affect and 0.885 for negative affect.

Flourishing Scale (FS) [12]. This scale is used to obtain information about how people evaluate their own flourishing, considered as a combination of feeling at ease and performing effectively, including positive relationships, feelings of competence and having meaning and purpose in life. It is composed of eight items that can be answered using a 7-point Likert scale (from 1-strongly disagree to 7-strongly agree). High scores indicate that respondents consider themselves to be on positive terms concerning important areas of functioning. It was validated in this general sample of Colombian adults and showed an internal consistency of 0.92 in the present sample [34].

Satisfaction with Life Scale (SWLS) [35]. This scale deals with the cognitive and overall assessment that a person has of their overall quality of life. It only contains five items answered with a 7-point Likert scale (from 1-strongly disagree to 7-strongly agree). Total scores range from 7 to 35, so higher scores indicate greater satisfaction with life. In this study, the Colombian version of this scale has been used [36]. Cronbach’s alpha in the sample of our study is 0.84.

Life Orientation Test-Revised (LOT-R). [37]. The scale is composed of 10 items: four control items, three that measure pessimism, and three that measure optimism. It uses a 5-point Likert scale (from 1-strong disagreement to 5-strong agreement). Higher scores in each subscale indicate high levels of optimism or pessimism, respectively. It has been validated in a general sample of Colombian adults showing good psychometric properties [38]. Cronbach’s alpha in our sample is 0.693 for Optimism and 0.636 for Pessimism. Although some authors question Cronbach’s Alpha values lower than 0.70, this consideration should not be taken as a “golden rule,” especially due to the reduced number of items on the LOT subscales, since an alpha that is too high could lead one to think that in reality, the three items measure the same indicator of the construct [38].

### 2.3. Data Analysis

Confirmatory factor analysis (CFA) was used to study the factorial structure of the SPANE. Different CFAs were calculated to test the single-factor, two-unrelated factor, and two-related factor models. The CFAs were first calculated for the version with item 9 (‘asustado’ (afraid)), and then for the version with item 13 (‘temeroso’ (afraid)). In this way, we attempted to check whether the result obtained in the qualitative study was the same as that obtained when studying the factorial structure.

A minimum cut-off of 0.90 for the comparative fit index (CFI), and a maximum cut-off of 0.08 for the root mean square error of approximation (RMSEA) and for the standard root mean square residual (SRMR) were considered as indicative of good fit [[39],[40],]. The factor measurement reliability [41] of the specified SPANE solution for each country was evaluated with the Composite Reliability Index (CRI) [42], which is identical to ω coefficient [43] because we have used the standardized factor loadings [44]. Then, the Average Variance Extracted (AVE) [45] was estimated to evidence factor measurement validity [44]. For the model that best fits the data, the corrected item-total polyserial correlations for the items in each subscale [46] have been calculated, as indicators of corrected homogeneity indices for items with ordinal response scales [47,48].

Gender-based measurement invariance was also studied for the best model, evaluated by calculating three nested invariance models that impose successive restrictions: configural, metric and scalar. According to Cheung Rensvold [49], when the sample size is adequate and similar between the different groups, a change of −0.010 or greater in CFI along with a change of 0.015 or greater in RMSEA, or a change of 0.030 or greater in SRMR would indicate that there is no invariance. To test residual invariance, a change of −0.010 or greater in CFI, together with a change of 0.015 or greater in RMSEA, or a change of 0.010 or greater in SRMR, would indicate that there is no invariance [50].

Convergent validity and concurrent validity have been studied using both models of structural equations, using the program Mplus 8.6 [51] with the robust maximum likelihood estimate (MLR). To study convergent validity, the items from the SPANE, the FS and the PANAS were considered. To study concurrent validity, the SPANE items were considered together with items of the Optimism and Pessimism subscales, as well as the SWLS items.

All the observed variables were regarded as ordinal, and parameters were calculated using maximum likelihood robust estimation (MLR). Some studies suggested the use of MLR when the data distributions are not normal, and if there are five or more response options to the items [52,53]. In these cases, a continuous distribution in the data can be assumed [54], while the variability of the calculated parameters is very small [55]. In addition, MLR typically makes less biased calculations of standard errors and presents good calculations of the correlations between the factors [56].

Finally, percentiles were calculated for both SPANE’s subscales, as well as for the balanced measure (SPANE-B) by sex. To calculate the descriptive data of the sample and obtain percentiles, the IBM SPSS 26 statistical package was used [57].

## 3. Results

### 3.1. Dimensionality and Item-Total Corrected Polyserial Correlations

The two-dimensional models with correlated factors (models 3 and 7) obtained the best fit indicators (see Table 1). However, since the models presented fairly high modification indexes between items 1 (positive) and 10 (contented), the fit was not considered to be so good in both cases. These residual covariances are theoretically justifiable, because when some persons consider that they are “often” very positive, it is reasonable that they also indicate that they are “often” contented. These adjectives refer to elements that capture feelings that are saturated with the valence dimension (pleasure/displeasure) of the emotion circumplex [12]. Furthermore, they refer to concepts that are semantically close (e.g., good/positive). In this case, positive is a general feeling and content is a more specific emotion related to that general feeling and, in some cultures, they are expected to be more closely interrelated [58,59]. For this reason, the models were calculated again, although we allowed for the correlation between the residuals of this item pair (models 4 and 8, respectively). These new models provided a better fit, especially that using item 13 ‘temeroso’ (afraid) (model 8). Although the RMSEA and CFI values are inconsistent, obtaining adequate values for RMSEA and inadequate (0.90) values for CFI sometimes occurs depending on the degrees of freedom of the model, that is, when there are enough degrees of freedom to obtain “good” values of RMSEA, but there are too few degrees of freedom to be able get “good” values from CFI [60]. Other authors indicate that the close fit tests based on the SRMR also yield acceptable type I error rates across all simulated conditions (always with ordinal responses, as in this case), regardless of the number of parameters to be estimated and the sample size. Compared to the RMSEA, the SRMR shows higher power in rejecting non-close fit models, especially in small samples (*n* ≤ 200). Therefore, the degree of misfit of an ordinal factor analysis model can be safely assessed using the SRMR [61]. For these reasons, we consider that model 8 shows a good fit to the data in this sample.

For this model (model 8), the AVE was good for negative affect (0.632) and positive affect (0.599). On the other hand, the CRI was also good for both subscales (0.801 for negative affect and 0.778 for positive affect). The remaining analyses were carried out with model 8, as it was shown to have the best fit (two correlated factors, with correlated errors between items 1 and 8, and with the adjective ‘temeroso’ (afraid)).

In Figure 1 the path diagram of model 8 is shown, with two correlated factors including item 13 (temeroso) (afraid) for the Colombian version of the SPANE instead of item 9 (asustado) (afraid). The standardized correlation between the latent variables (positive and negative factors) was −0.592. The factor loadings were statistically significant (*p* < 0.001), ranging from 0.323 to 0.790 for the SPANE-P, and from 0.464 to 0.712 for the SPANE-N. Item-total corrected polyserial correlations ranged from 0.622 to 0.771 (*SE* from 0.014 to 0.020) for the SPANE-P, and from 0.472 to 0.614 for the SPANE-N (*SE* from 0.016 to 0.026).

### 3.2. Gender Measurement Invariance

In Table 2 the results for the gender-based measurement invariance models are shown. The model’s fit indices in the groups of men and women are acceptable. The results of the invariance models show that there is configural and metric invariance, but not scalar, in view of its ΔCFI 0.10. Therefore, the intercepts were reviewed, finding that items 8 (‘triste’) (sad) and 11 (‘enfadado’) (angry) demonstrated an important difference between men and women. The intercepts of these items were set at 0 for the group of women, and satisfactory results were obtained from the partial scalar invariance. The latent mean values were set to zero for men. The results showed that there were no differences between genders in positive (*b* = −0.044, *z* = −1.055, *p* = 0.291) or negative affects (*b* = 0.076, *z* = 1.584, *p* = 0.113).

### 3.3. Convergent and Concurrent Validity

To study validity, model 8 was fitted to data. The convergent validity model showed acceptable fit to data: χ^2^(729) = 2698.486 (*p* < 0.001), CFI = 0.889, RMSEA = 0.046, RMSEA 90% CI (0.045, 0.048), and SRMR = 0.059. However, two modification indices (MI) between residuals showed high values: the MI for items 7 (scared) and 20 (afraid) of the PANAS (*MI* = 134.272), and item 13 (afraid) of the SPANE and 20 (afraid) of the PANAS (MI = 113.977). Again, these residual covariances are theoretically justifiable, because they refer to pairs of elements that capture feelings that are either same (“scared” and “afraid”). For this reason, model 8 was adjusted again, but estimating the residual correlation between these pairs of items, showing now a very good fit: χ^2^(727) = 2496.787 (*p* < 0.001), CFI = 0.901, RMSEA = 0.044, RMSEA 90% CI [0.042, 0.046], and SRMR = 0.058. Factor loadings for this model were statistically significant (*p* < 0.001) and in the expected sense, ranging from 0.615 to 0.842 for the FS, from 0.601 to 0.781 for the SPANE-P, from 0.523 to 0.709 for the SPANE-N, from 0.165 to 0.770 for the PANAS-P and from 0.495 to 0.709 for the PANAS-N. In Figure 2 the path diagram for this model can be seen.

On the other hand, the concurrent validity model (see Figure 3) carried out also showed very good fit to data: χ^2^(219) = 738.99 (*p* < 0.001), CFI = 0.936, RMSEA = 0.044, RMSEA 90% CI (0.040, 0.047), and SRMR = 0.040. All factor loadings were statistically significant (*p* < 0.001) and as expected, ranging from 0.606 to 0.834 for the SWLS, from 0.554 to 0.774 for Optimism, from 0.497 to 0.807 for Pessimism, from 0.586 to 0.780 for SPANE-P and from 0.476 to 0.723 for the SPANE-P. In Table 3 the correlations among latent variables for the convergent and concurrent validity models are shown.

A Kolmogorov–Smirnov test indicated that none of the SPANE-P (Z = 3.579, *p* < 0.001), SPANE-N (*z* = 2.099, *p* < 0.001) or SPANE-B (*z* = 2.141, *p* < 0.001) scales does not fit a normal distribution. Therefore, norms for both subscales in terms of percentiles are presented in Table 4 by gender.

## 4. Discussion

In the framework of the study of well-being from the tripartite model proposed by Diener, the balance of self-evaluation of satisfaction with life and positive and negative feelings is included. In this sense, the SPANE is an instrument aimed at offering information about the frequency with which people consider that they experience positive or negative emotions, rather than the intensity with which they experience them. This study aimed at adjusting the SPANE scale to the general Colombian population and studying its psychometric properties, in order to introduce it as an assessment tool in normal practice. As a result, first, the objective was to ensure the psycholinguistic equivalence of the items through an iterative purification process to achieve a culturally appropriate version suitable for the Colombian population. The current recommendations of the International Test Commission for test adaptation purposes were followed, in addition to the compliance criteria checklist that was recently proposed by Hernández et al. [32].

Thus, the Spanish versions of the items which were developed with the back-translation procedure were selected, as recommended [33,34], but we also considered the criteria outlined by judges with a knowledge of the Colombian cultural context. This procedure made it possible to identify that the adjective ‘asustado’ (afraid) in the Spanish version had a different meaning for Colombians, so this word was replaced with ‘temeroso’ (afraid). It should be noted that the need for this substitution was established by taking not only the judges’ assessments into account, but also the opinions of the people who participated in the qualitative phase of this research, in addition to the results of the statistical analyses, which showed a better fit of the models when this adjective was used. The final version of the adjectives in the scale for the Colombian population is shown in Appendix A, including the item ‘’Temeroso’ instead of ‘Asustado.’ It was also possible to confirm that the test instructions were understood correctly, and that the administration method did not affect its comprehension (i.e., paper-and-pencil versus computer-administered test). As several authors have asserted [29,30], publications on adapting measurement instruments that address these aspects are scarce, even though they are considered essential by international standards [31]. Therefore, this is a contribution of the present study.

On the other hand, a large general population sample has been used, from different regions of Colombia, with different occupations and educational levels, which allows us to provide preliminary normative data, which are offered for men and women considering that the scale does not have gender-based invariance in our sample. In addition, percentiles are recommended for use as a reference in order to rate the scale, due to the lack of data normality, which makes it impossible to use for the purposes of calculating the rarity of an individual’s score [62].

As in studies conducted in various countries and contexts [14,16,22,24,25,27], the bi-factorial structure of the scale was corroborated. While the CFI value does not reach the cut-off point of 0.90, the RMSEA and SRMR indices do have values indicating an acceptable fit. Even though some authors indicate that the CFI must be greater than 0.90 to be considered a good fit, others state that these are arbitrary values, so comparing the CFI of different models is recommended, considering higher values as the best [63,64]. Like Li et al. [25], the model’s fit of two correlated factors was found to improve when errors were correlated, which indicates that the SPANE-P and SPANE-N scales show two different factors, which are moderately negatively correlated, when the measurement error is controlled. In other words, positive and negative feelings are clearly separable although not orthogonal. This is consistent with previous views on the separate measurement of positive and negative emotional well-being [7,65,66] and the perspective of Diener et al. [12] on the development of SPANE. We agree with Li [25] that this does not pose any problem in the analysis since, in our case, only the correlation between residuals of items 1 (positive) and 10 (joyful) was completed. Therefore, out of a total of 66 potentially correlated errors, only two were allowed. In addition, the correlation between residuals did not substantially change the values of the factor loadings or the correlation between SPANE-P and SPANE-N.

It is also worth noting that the Average Variance Extracted Index was good for both negative and positive affect, with values above the minimum level of 0.50, as well as the Composite Reliability Index (over 0.75 for both subscales), data supporting the validity and reliability of the scale for Colombia. Convergent validity was also demonstrated with PANAS, an instrument that has been widely used to study positive and negative affectivity in the Colombian population [67,68], and criteria validity indicators were obtained with other measures of subjective well-being (such as flourishing and satisfaction with life) and personality (such as optimism and pessimism), consistent with that reported in the abundant scientific literature on the components of subjective well-being [68,69,70] and its determinants [71,72,73]. It is worth emphasizing that SPANE is a good measure of the affective component of well-being. However, researchers who wish to assess the intensity or other qualities of affectivity (such as arousal) should consider other instruments.

The assessment of gender-based invariance is important because it indicates the extent to which the construct being assessed is understood in the same way by both men and women [74]. Our results indicate that certain items are not interpreted by men and women in the same way, which differ from those reported by Espejo et al. [29] who found scalar invariance in the Spanish sample, just as Li [25] did with the Chinese population. Jovanovic [22] found partial gender-based invariance among the Serbian population, since some items were not invariant. However, our results coincide with those reported by Daniel-González [27] for the Mexican population, which can be explained by the cultural similarities between both countries, in which a macho culture predominates [75,76]. In this sense, interestingly, it was precisely the sad and angry adjectives which presented the greatest differences between men and women, since in the Latin American imagination, emotions such as sadness and fear are considered more ‘feminine’ while anger is more attributed to be ‘masculine’ [77].

## 5. Limitations and Future Directions

One of the limitations of this study is the impossibility of obtaining a national representative sample. We resorted to availability sampling, which limits the ability to generalize the results, especially considering Colombia’s cultural diversity. In addition, administering the test online is limited to the cultural background of the participants. In this sense, we believe that the psychometric properties of the scale should be studied in other populations, for example, rural residents. Additionally, future studies should include a greater number of older people, taking into account that emotional regulation changes significantly throughout life and that appropriate and relevant methods for young people may not be suitable for middle or older adults [5]. Although the present study included participants up to 67 years of age, younger participants predominated, among other reasons because the application was exclusively online.

Another limitation is that we did not examine the temporal stability or sensitivity to detect changes in the affectivity of the SPANE. Thus, for future studies, we recommend obtaining test–retest comparisons, as well as the use of the scale in the clinical population, in order to detect changes in the experiences of the affects related to various psychopathological disorders.

It will also be enlightening to carry out a comparative analysis based on other demographic variables such as age, socioeconomic status, and academic background. Likewise, it would be advisable to study cross-cultural invariance, which would help to better understand the cultural, socioeconomic, and demographic determinants of affectivity.

## 6. Conclusions

At a theoretical level, our data confirmed that the SPANE is consistent with the statement of its authors regarding the non-orthogonal bi-factorial structure of negative and positive affectivity as indicators of the hedonic dimension of well-being, coinciding with that reported by studies carried out in other countries. From the practical point of view, this study showed that the Scale of Positive and Negative Experience (SPANE) is useful in developing new research in the fields of positive psychology and health psychology in Colombia, especially considering, in terms of convenience, its ease of understanding, application and interpretation, which allow for its use in multiple fields of psychological practice and research.

Second, our study found that the general latent structures (the number of factors and the pattern of factor-element relationships) of this measure of well-being are identical between men and women (configural invariance) and that it is also reasonable to conclude that the items are linked in the same way to the factors, which indicates that these coefficients are comparable (metric invariance). However, scalar invariance was not confirmed for all items, which should alert researchers to the risk of comparing affectivity average scores between both groups. It would be more appropriate to compare it on an item-by-item basis for each sex, since not all items seem to be perceived in the same way by both groups. Identifying these differences instead of continuing to assume the invariance of theoretically non-invariant measures or indicators is risky, as it can lead to erroneous inferences with potential implications for taking action and developing policies with a gender perspective, reinforcing erroneous sexist beliefs. For this reason, we provide percentiles by sex, to facilitate the interpretation of the results of future studies in Colombia.

## Figures and Tables

**Figure 1 ijerph-18-06449-f001:**
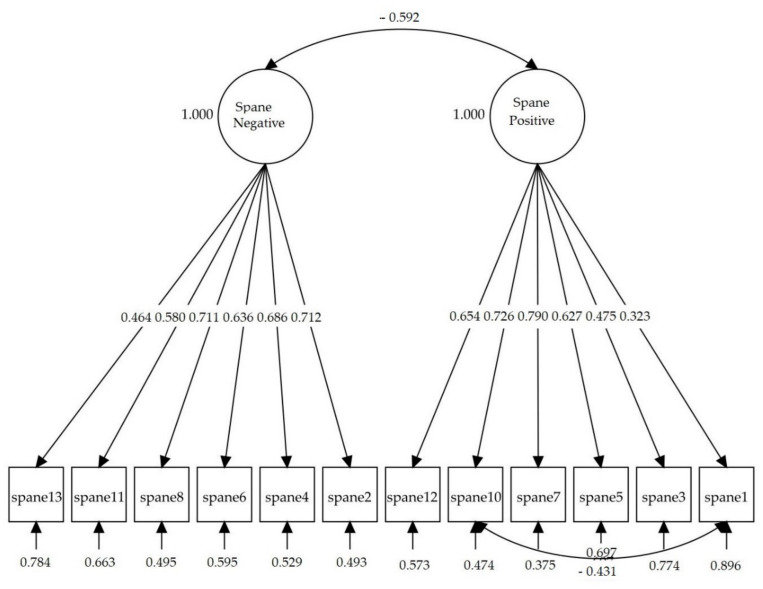
Path diagram and standardized loadings of model 8 for the Scale of Positive and Negative Experience (SPANE), with two correlated factors and item 13 (temeroso)(afraid) replacing item 9 (asustado) (afraid).

**Figure 2 ijerph-18-06449-f002:**
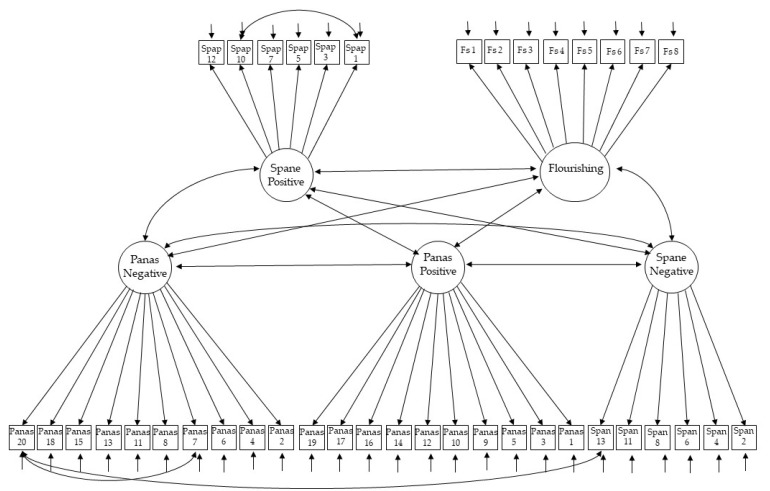
Path diagram for the convergent validity model. Panas: Positive and Negative Affect Schedule; Spane: Scale of Positive and Negative Experience.

**Figure 3 ijerph-18-06449-f003:**
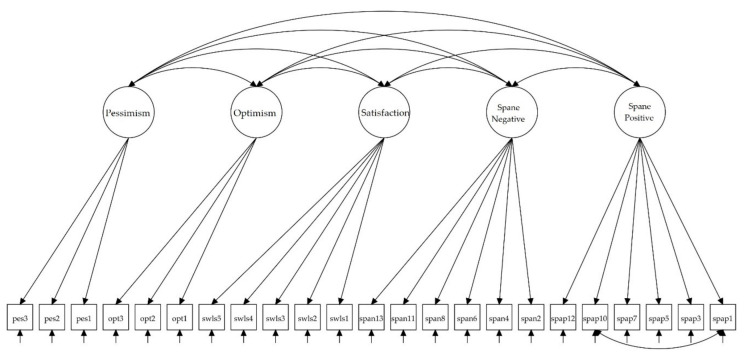
Path diagram for the concurrent validity model. Spane: Scale of Positive and Negative Experience.

**Table 1 ijerph-18-06449-t001:** Goodness of fit of the confirmatory models tested for the Scale of Positive and Negative Experience (SPANE).

Models with Item 9 (Asustado/Afraid)	χ^2^	df	CFI	RMSEA	RMSEA 90% CI	SRMR
(1) Single factor	839.006 *	54	0.629	0.108	0.102, 0.115	0.089
(2) 2 independent factors (Spane Positive and Spane Negative)	627.715 *	54	0.729	0.093	0.086, 0.099	0.158
(3) 2 correlated factors (Spane Positive and Spane Negative)	453.675 *	53	0.811	0.078	0.072, 0.085	0.061
(4) 2 correlated factors (Spane Positive and Spane Negative) with correlated errors ^a^	357.058 *	51	0.856	0.069	0.067, 0.076	0.054
Models with item 13 (temeroso/afraid)						
(5) Single factor	795.602 *	54	0.647	0.105	0.099, 0.112	0.085
(6) 2 independent factors (Spane Positive and Spane Negative)	596.662 *	54	0.742	0.090	0.084, 0.097	0.159
(7) 2 correlated factors (Spane Positive and Spane Negative)	416.587 *	53	0.827	0.074	0.068, 0.081	0.057
(8) 2 correlated factors (Spane Positive and Spane Negative) with correlated error ^a^	319.375 *	51	0.872	0.065	0.058, 0.072	0.044

Note. df = degrees of freedom; CFI = comparative fit index; RMSEA = Root-Mean-Square error of approximation; CI = confidence interval; SRMR = standardized Root-Mean-Squared residual. * *p* < 0.001. ^a^ Correlated error for item 1 (positive) and item 10 (joyful).

**Table 2 ijerph-18-06449-t002:** Gender-Based Measurement Invariance Models of the Scale of Positive and Negative Experience (SPANE) (Reference Group: Men).

Model 8	χ^2^	df	Δχ^2^	Δgl	CFI	RMSEA	SRMR	ΔCFI	ΔRMSEA	ΔSRMR
Men	135.337 *	52			0.914	0.060	0.046			
Women	283.159 *	52			0.896	0.075	0.046			
Configural	416.279 *	104	-	-	0.914	0.070	0.046	-	-	-
Metric	434.986 *	114	16.911	10	0.912	0.067	0.054	−0.002	−0.003	0.008
Scalar	495.265 *	124	67.338	10	0.898	0.069	0.058	−0.014	0.004	0.004
Partial Scalar **	463.893 *	122	28.383	8	0.906	0.067	0.057	−0.006	0.000	0.003

Note. df = degrees of freedom; Δχ^2^ = Chi Square increase; Δgl = degrees of freedom increase; CFI = comparative fit index; RMSEA = Root-Mean-Square error of approximation; ΔCFI = CFI increase; ΔRMSEA = RMSEA increase. * *p* < 0.001. ** Intercepts of the items 8 and 11 fixed to 0.

**Table 3 ijerph-18-06449-t003:** Correlation coefficients (standard errors) between the Scale of Positive and Negative Experience (SPANE) subscales and well-being measures in the validity models.

	Convergent Validity				Concurrent Validity			
	SPANE Positive Experience	SPANE Negative Experience	PANAS Positive Affect	PANAS Negative Affect	SPANE Positive Experience	SPANE Negative Experience	Satisfaction	Optimism
Flourishing	0.516 (0.042)	−0.286 (0.038)	0.466 (0.035)	−0.222 (0.036)				
PANAS Negative Affect	−0.463 (0.035)	0.774 (0.022)	−0.250 (0.042)					
PANAS Positive Affect	0.739 (0.026)	−0.437 (0.039)						
SPANE Negative Experience	−0.614 (0.036)							
SPANE Negative Experience					−0.614 (0.036)			
Satisfaction					0.708 (0.025)	−0.453 (0.034)		
Optimism					0.759 (0.027)	−0.568 (0.037)	0.732 (0.029)	
Pessimism					−0.314 (0.042)	0.496 (.037)	−0.240 (0.043)	−0.257 (0.051)

Note. PANAS = Positive and Negative Affect Schedule. SPANE: Scale of Positive and Negative Experience. All correlations were statistically significant (*p* < 0.001).

**Table 4 ijerph-18-06449-t004:** Percentiles for the Scale of Positive and Negative Experience (SPANE), and for the balanced measure.

	Male			Female		
Percentiles	SPANE Positive	SPANE Negative	SPANE Balanced	SPANE Positive	SPANE Negative	SPANE Balanced
5	18	7	−1	17	9	−2
10	18	9	0	19	10	0
15	20	10	2	20	11	2
20	21	11	3	21	12	3
25	22	11	5	22	12	5
30	23	12	6	22	13	6
35	23	12	7	23	13	7
40	24	13	8.8	24	14	8
45	24	13	10	24	14	9
50	25	14	10	24	15	10
55	25	15	11	25	15	10
60	25	15	12	25	16	11
65	26	16	14	26	16	12
70	26	17	14	26	17	13
75	27	17	15	27	18	13
80	28	18	16	27	18	14
85	28	18	17	28	19	16
90	29	19	19	29	20	17
95	30	21	22	30	22	19

## Data Availability

Data sharing not applicable.

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
