# Peer review of "Adaptation and Psychometric Properties of the Scale of Positive and Negative Experience (SPANE) in the General Colombian Population"

_ijerph, 2021, doi:10.3390/ijerph18126449_

Round 1

Reviewer 1 Report

I would recommend to put more attention to the limitations of the study. Besides the limitations named by authors, the sample is mainly from young people - although thei claim that the participants were 18-67, the mean 25 suggests that there were only few participants over 40. Meanwhile, the emotional regulation changes significantly over lifespan and methods adequate and relevant for young people may not be suitable for middle or older adults.

Author Response

Reviewer 1

I would recommend to put more attention to the limitations of the study. Besides the limitations named by authors, the sample is mainly from young people - although they claim that the participants were 18-67, the mean 25 suggests that there were only few participants over 40. Meanwhile, the emotional regulation changes significantly over lifespan and methods adequate and relevant for young people may not be suitable for middle or older adults.

Answer: Thank you very much for your kind comments. Your suggestion has been added to the discussion, on line 430.

Reviewer 2 Report

  • The article meets all the requirements for publication.
  • It provides a new instrument to evaluate the affective component of well-being in Colombia. Future studies of emotional well-being in Colombia may benefit from the use of this scale.
  • The evaluation of well-being is very useful for the intervention in prevention and health promotion.
  • From a psychometric point of view, it meets the criteria of the International Testing Commission for the adaptation of a test.
  • Adapting an instrument to different countries facilitates intercultural research.
  • For all these reasons, I recommend its publication.

Author Response

Reviewer 2

Comments and Suggestions for Authors

  • The article meets all the requirements for publication.
  • It provides a new instrument to evaluate the affective component of well-being in Colombia. Future studies of emotional well-being in Colombia may benefit from the use of this scale.
  • The evaluation of well-being is very useful for the intervention in prevention and health promotion.
  • From a psychometric point of view, it meets the criteria of the International Testing Commission for the adaptation of a test.
  • Adapting an instrument to different countries facilitates intercultural research.
  • For all these reasons, I recommend its publication.

Answer: Thank you very much for your kind comments.

Reviewer 3 Report

The manuscript has been well-written. However, there are major methodological issues need to be addressed by the authors.

1) Please provide the page number of the direct quotation from Diener’s SPANE (p. 2) or paraphrase the sentences.

2) LOT-R with alpha value lower than the acceptable range > 0.70 (p. 4). Please highlight it in the limitation and evaluate whether it may affect the results related to the concurrent validity of the scale.

3) The major issue in this study is that, the CFA results show that the Colombian version of SPANE with problematic factorial validity (Table 1), none of the models fulfil the basic criteria for adequate model fit, i.e. CFI > 0.090. The authors attempted to correlating the error terms, but still failed to fulfil the criteria. The authors also need to note that theoretical justifications should be provided if correlating the error terms (Hermida, 2015).

4) The authors need to provide justifications from the literature for the use of SEM to evaluate the concurrent validity of the SPANE together with PANAS and LOT-R. Most of the scale validation literature would suggest to use correlation for evaluating the concurrent validity with other construal related scale and its sub-scales. The authors reported the SEM results of the concurrent validity may confuse the readers that the scale with good factorial validity. I think the entire section need to be re-written to avoid any confuses to the reader.

References

Hermida, R. (2015). The problem of allowing correlated errors in structural equation modeling: concerns and considerations. Computational Methods in Social Sciences (CMSS), 3(1), 05-17. Retrieved from https://EconPapers.repec.org/RePEc:ntu:ntcmss:vol3-iss1-15-005

Author Response

Reviewer 3

Comments and Suggestions for Authors

The manuscript has been well-written. However, there are major methodological issues need to be addressed by the authors.

1) Please provide the page number of the direct quotation from Diener’s SPANE (p. 2) or paraphrase the sentences.

2) LOT-R with alpha value lower than the acceptable range > 0.70 (p. 4). Please highlight it in the limitation and evaluate whether it may affect the results related to the concurrent validity of the scale.

3) The major issue in this study is that the CFA results show that the Colombian version of SPANE with problematic factorial validity (Table 1), none of the models fulfil the basic criteria for adequate model fit, i.e. CFI > 0.090. The authors attempted to correlating the error terms, but still failed to fulfil the criteria. The authors also need to note that theoretical justifications should be provided if correlating the error terms (Hermida, 2015).

4) The authors need to provide justifications from the literature for the use of SEM to evaluate the concurrent validity of the SPANE together with PANAS and LOT-R. Most of the scale validation literature would suggest using correlation for evaluating the concurrent validity with other construal related scale and its sub-scales. The authors reported the SEM results of the concurrent validity may confuse the readers that the scale with good factorial validity. I think the entire section need to be re-written to avoid any confuses to the reader.

Answer: Thank you very much for your kind comments. Regarding the questions that you indicate:

  • The page number has been added.
  • Although the alpha value of the Pessimism subscale of the LOT is less than .70, it can be considered adequate since it is a factor composed of only three items. If it were quite a few more items it could be problematic, but since there are so few items, it is not that much. Furthermore, perhaps if it were too high, one could think that, in reality, at least two of the three items could be measuring the same indicator of the Pessimism construct (Ventura-León & Peña-Calero, 2020) (lines 198-202).
  • It is true that in this case the CFI value does not reach the cutoff point of .90, established as adequate. However, the RMSEA and SRMR values are below the cut-off points established to consider that the model presents a good fit. These two fit indices are indicators of the error offered by the model, that is, they indicate the degree of discrepancy between the data and the model. On the other hand, CFI proportionally describes how well the researcher's model explains the covariance matrix above and beyond the baseline model (generally the null model, which assumes that all observed variables are independent).

According to Lai & Green (2016), RMSEA and CFI can provide inconsistent fit assessments under certain conditions, although this inconsistency is not necessarily a diagnosis of problems in the model specification or in the data. It may occur perhaps because the two indices, due to how they are constructed, evaluate the fit from different perspectives. When RMSEA and CFI are inconsistent, it is not necessary to automatically ignore the model just because an index does not meet the limit, but neither should the model be retained by reporting only the index that provides good fit. An attempt should be made to explain why the indices disagree and the implications of this disagreement. Obtaining adequate values of RMSEA and inadequate (<.90) of CFI sometimes occurs depending on the degrees of freedom of the model, that is, when there are enough degrees of freedom to obtain "good" values of RMSEA but there are too few degrees of freedom to be able get "good" values from CFI (Lai & Green, 2016). This is what happens with our models in Table 1 (around 51 df). However, the models for convergent and concurrent validity, with many more degrees of freedom, offer adequate values for both fit indices, and they include the SPANE bi-factor structure. Although these authors (Lai & Green, 2016) perform a simulation study with data that are normally distributed, it is easy for RMSEA and CFI disagreements to occur, which can cause confusion, but they are unlikely to be indicators of problems of adjustment. Other authors indicate that the close fit tests based on the SRMR also yield acceptable type I error rates across all simulated conditions (always with ordinal responses, as in this case), regardless of the number of parameters to be estimated and the sample size. Compared to the RMSEA, the SRMR shows higher power to reject non-close fit models, especially in small samples (N ≤ 200). Therefore, the degree of misfit of an ordinal factor analysis model can be safely assessed using the SRMR (Shi et al., 2020). For these reasons, we consider that the model 8 shows a good fit to the data in this sample.

It has been very interesting for us to delve into this question, and we have made a brief reference to the possible reasons for the inconsistency between the RMSEA and CFI values in the text (line 259).

Lai, K., & Green, S. B. (2016). The problem with having two watches: Assessment of fit when RMSEA and CFI disagree. Multivariate behavioral research, 51(2-3), 220-239. https://doi.org/10.1080/00273171.2015.1134306

Shi, D., Maydeu-Olivares, A., & Rosseel, Y. (2020). Assessing fit in ordinal factor analysis models: SRMR vs. RMSEA. Structural Equation Modeling: A Multidisciplinary Journal27(1), 1-15. https://doi.org/10.1080/10705511.2019.1611434

Regarding the correlation between errors, this can sometimes occur when the indicator variables share components, as Hermida says. These residual covariances are theoretically justifiable, because they refer to pairs of elements that capture feelings that are either same ("positive" and "content", “scared” and “afraid”, “afraid” of the SPANE and “afraid” of the PANAS) or opposite in valence. General feelings are saturated with the valence dimension (pleasure/displeasure) of the emotion circumplex (Diener et al., 2010) and have similar arousal levels. Furthermore, they refer to concepts that are semantically close (e.g., good/positive, bad/negative) or antonyms (e.g., good/bad, positive/negative). In this case, positive is a general feeling and content is a more specific emotion of that general feeling and, in some cultures, they are expected to be more closely interrelated among them (Scherer et al., 2013; Yik et al., 2011).

This topic has been included in the article, on line 249 and on line 314.

Diener, E., Wirtz, D., Tov, W. et al. New Well-being Measures: Short Scales to Assess Flourishing and Positive and Negative Feelings. Social Indicators Research, 97, 143–156 (2010). https://doi.org/10.1007/s11205-009-9493-y

Scherer, K. R., Shuman, V., Fontaine, J. R. J., & Soriano, C. (2013). The GRID meets the wheel: Assessing emotional feeling via self-report. In J. R. J. Fontaine, K. R. Scherer, & C. Soriano (Eds.), Components of emotional meaning: A sourcebook (pp. 281–298). Oxford University Press.

Yik, M., Russell, J. A., & Steiger, J. H. (2011). A 12-point circumplex structure of core affect. Emotion, 11, 705–731. https://doi.org/10.1037/a0023980

  • Regarding the models to study validity, if the correlation between the total scores of the different measures used to study validity is calculated, correlations will be obtained between observed scores, which contain error, and the obtained indicators will underestimate validity. If the correlation between the latent variables is calculated using an SEM, the error will be extracted from these correlations, which is estimated separately, as occurs when estimating a CFA. In this way, a much more reliable estimate of the relationship between the measurements is obtained. SEMs become a way to estimate validity much more powerful and closer to reality. For this reason, no correlations have been estimated between the total scores in the measures considered. This explanation has been included in the introduction so that the reader better understands the reason for using this methodology, on line 107.

Reviewer 4 Report

The authors tested the psychometric properties of the SPANE in a Colombian sample. I am not aware of any studies that have done this, so if this is new, I think it's a worthwhile endeavor. The sample seems reasonable and I thought the authors did a reasonable job of conducting the study. 

My main concern is that the SPANE is not a very good measure of affect, in my opinion. The authors rightly critique the PANAS, but I do not believe the SPANE is much better. The SPANE fails to distinguish arousal, an important topic particularly in cross-cultural research. I also find the use of broad, general words (e.g., positive) to be a weakness. The goal of measuring affect is to study specific emotional states, but yet broad descriptive words like positive merely tap into general evaluations of life, such as life satisfaction. This muddies the conceptual water a bit. However, given that many people use the SPANE, translating the items and testing for psychometric properties in a Colombian sample seems reasonable. I would recommend that the authors describe some of the limitations/weaknesses of the SPANE in the discussion. 

Perhaps I missed this, but what time frame were participants instructed to think about as they answered the SPANE adjectives? One issue I have with lengthy retrospective questionnaires is that biases and heuristics may influence their judgments. Thus, the measure may not accurately assess how frequently they experience positive and negative affective states. Rather, it assesses broad views about how people view themselves. I think it would be helpful to mention this in the discussion. 

I think the CFI value of .889 is a bit low. I would not describe this as adequate.  

Author Response

Reviewer 4

The authors tested the psychometric properties of the SPANE in a Colombian sample. I am not aware of any studies that have done this, so if this is new, I think it's a worthwhile endeavor. The sample seems reasonable, and I thought the authors did a reasonable job of conducting the study. 

My main concern is that the SPANE is not a very good measure of affect, in my opinion. The authors rightly critique the PANAS, but I do not believe the SPANE is much better. The SPANE fails to distinguish arousal, an important topic particularly in cross-cultural research. I also find the use of broad, general words (e.g., positive) to be a weakness. The goal of measuring affect is to study specific emotional states, but yet broad descriptive words like positive merely tap into general evaluations of life, such as life satisfaction. This muddies the conceptual water a bit. However, given that many people use the SPANE, translating the items and testing for psychometric properties in a Colombian sample seems reasonable. I would recommend that the authors describe some of the limitations/weaknesses of the SPANE in the discussion. 

Perhaps I missed this, but what time frame were participants instructed to think about as they answered the SPANE adjectives? One issue I have with lengthy retrospective questionnaires is that biases and heuristics may influence their judgments. Thus, the measure may not accurately assess how frequently they experience positive and negative affective states. Rather, it assesses broad views about how people view themselves. I think it would be helpful to mention this in the discussion. 

I think the CFI value of .889 is a bit low. I would not describe this as adequate.  

Answer: Thank you very much for your kind comments.

Regarding the construct evaluated by SPANE, this questionnaire assesses broad points of view on how people see themselves. It does not evaluate affectivity, but positive and negative experiences, as Diener himself indicates in the text quoted literally: "By including labels such as' 'good' 'and' 'positive' ', and' 'bad' 'and' 'negative' ', that reflect all types of feelings, the SPANE assesses the full range of possible desirable and undesirable experiences". In the discussion we have added, to highlight this issue, that to assess the intensity or other qualities of affectivity (such as arousal), other measurement instruments should be considered (line 407).

Regarding the fit indices, it is true that in this case the CFI value does not reach the cutoff point of .90, established as adequate. However, the RMSEA and SRMR values are below the cut-off points established to consider that the model presents a good fit. These two fit indices are indicators of the error offered by the model, that is, they indicate the degree of discrepancy between the data and the model. On the other hand, CFI proportionally describes how well the researcher's model explains the covariance matrix above and beyond the baseline model (generally the null model, which assumes that all observed variables are independent). According to Lai & Green (2016), RMSEA and CFI can provide inconsistent fit assessments under certain conditions, although this inconsistency is not necessarily a diagnosis of problems in the model specification or in the data. It may occur perhaps because the two indices, due to how they are constructed, evaluate the fit from different perspectives. When RMSEA and CFI are inconsistent, it is not necessary to automatically ignore the model just because an index does not meet the limit, but neither should the model be retained by reporting only the index that provides good fit. An attempt should be made to explain why the indices disagree and the implications of this disagreement. Obtaining adequate values of RMSEA and inadequate (<.90) of CFI sometimes occurs depending on the degrees of freedom of the model, that is, when there are enough degrees of freedom to obtain "good" values of RMSEA but there are too few degrees of freedom to be able get "good" values from CFI (Lai & Green, 2016). This is what happens with our models in Table 1 (around 51 df). However, the models for convergent and concurrent validity, with many more degrees of freedom, offer adequate values for both fit indices, and they include the SPANE bi-factor structure. Although these authors (Lai & Green, 2016) perform a simulation study with data that are normally distributed, it is easy for RMSEA and CFI disagreements to occur, which can cause confusion, but they are unlikely to be indicators of problems of adjustment.

It has been very interesting for us to delve into this question, and we have made a brief reference to the possible reasons for the inconsistency between the RMSEA and CFI values in the text (line 259).

Lai, K., & Green, S. B. (2016). The problem with having two watches: Assessment of fit when RMSEA and CFI disagree. Multivariate behavioral research, 51(2-3), 220-239. https://doi.org/10.1080/00273171.2015.1134306

Round 2

Reviewer 3 Report

Thanks for addressing my concerns. I am fully satisfied with the responses and changes made by the authors.